# EMBODIEDCITY: A BENCHMARK PLATFORM FOR EMBODIED AGENT IN REAL-WORLD CITY ENVIRONMENT

## ABSTRACT

Embodied artificial intelligence (EmbodiedAI) emphasizes the role of an agent's body in generating human-like behaviors. The recent efforts on EmbodiedAI pay a lot of attention to building up machine learning models to possess perceiving, planning, and acting abilities, thereby enabling real-time interaction with the world. However, most works focus on bounded indoor environments, such as navigation in a room or manipulating a device, with limited exploration of embodying the agents in open-world scenarios. That is, embodied intelligence in the open and outdoor environment is less explored, for which one potential reason is the lack of high-quality simulators, benchmarks, and datasets. To address it, in this paper, we construct a benchmark platform for embodied intelligence evaluation in real-world city environments. Specifically, we first construct a highly realistic 3D simulation environment based on the real buildings, roads, and other elements in a real city. In this environment, we combine historically collected data and simulation algorithms to conduct simulations of pedestrian and vehicle flows with high fidelity. Further, we designed a set of evaluation tasks covering different EmbodiedAI abilities. Moreover, we provide a complete set of input and output interfaces for access, enabling embodied agents to easily take task requirements and current environmental observations as input and then make decisions and obtain performance evaluations. On the one hand, it expands the capability of existing embodied intelligence to higher levels. On the other hand, it has a higher practical value in the real world and can support more potential applications for artificial general intelligence. Based on this platform, we evaluate some popular large language models for embodied intelligence capabilities of different dimensions and difficulties. The executable program of this platform is available for download, and we have also released an easy-to-use Python library and detailed tutorial documents. All of the software, Python library, codes, datasets, tutorials, and real-time online service are available on this anonymous website: `https://embodied-ai.city`.

## 1 INTRODUCTION

Embodied AI (Duan et al., 2022) serves as the recent advance of artificial intelligence, presenting an emerging paradigm shift from the traditional artificial intelligence, which learns from static datasets (e.g., ImageNet, which contains 2D images). Specifically, embodied artificial intelligence is expected to behave like a real human, which is able to learn from the environment and dynamically interact with the world, considered an essential approach to Artificial General Intelligence (AGI) (Duéñez-Guzmán et al., 2023). Various tasks for embodied intelligence have been established in different domains, including robotics (He et al., 2023; Barreiros et al., 2022; Driess et al., 2023), game AI (Fan et al., 2022a; Nottingham et al., 2023), unmanned vehicles/aerial drones (Zhou et al., 2022), etc. Generally speaking, EmbodiedAI requires the agent to accurately understand the environment, perform high-level reasoning, and effectively choose appropriate actions to execute tasks. Therefore, the training and testing of embodied intelligence models are closely related to the environment. To accelerate training efficiency and test embodied agents more conveniently, researchers generally choose to construct a simulation environment as an approximation and mirror of the real world. Specifically, the key to this process is providing an environment where embodied agents can obtain observations in real-time from a first-person perspective, generate actions, and receive feedback (Padalkar et al.,

Figure 1: An embodied environment built upon real cities includes realistic urban landscapes such as streets, buildings, city elements, pedestrians, and traffic. It also offers interaction interfaces for aerial and ground agents with the urban environment.

2023). This environment should further support the practical implementation and application of embodied agents.

However, the environments and tasks for embodied intelligence the existing research mainly focuses on are relatively limited (Duan et al., 2022). Many works (Shridhar et al., 2020; Deitke et al., 2022; Gao et al., 2021) only consider the indoor scenarios for embodied intelligence. These tasks include visual QA tasks targeting certain objects or simple task decomposition in the room (Azuma et al., 2022; Francis et al., 2022). Such benchmarks actually restrict the validation of embodied agents' capabilities within a very narrow boundary, with low task difficulty and a large gap to artificial general intelligence. Therefore, in this paper, we consider extending embodied agents from indoor rooms to outdoor cities, expanding tasks beyond the indoor spaces to a broader urban environment.

There are a few works that build a 3D urban environment for EmbodiedAI such as MetaUrban (Wu et al., 2024), GRUtopia (Wang et al., 2024), and AerialVLN (Liu et al., 2023), but they are limited in either the environment or the benchmark (**we analyze the limitations in detail from 9 dimensions in Table 1**). Specifically, all of these works consider a fictional city. Either they employ a highly simplified environment or only set up one or two embodied tasks. There are also some other efforts that use street view images to construct the environment, which significantly limits the potential Embodied AI tasks. To address the limitations, in this work, we first build a simulator for a highly realistic 3D simulation environment of a city, as shown in Figure 1. The environment is based on the real streets, buildings, city elements, pedestrians, and traffic in one commercial district from one of China's largest cities, Beijing. In this commercial district, we establish realistic and detailed city-building 3D models as the foundation for the entire benchmark platform. Furthermore, we combine the historically collected real-world traffic data and simulation algorithms to conduct simulations of pedestrian and vehicle flows. We then set up a set of evaluation tasks covering different types of embodied capabilities for embodied intelligence, including scene description, question answering, dialogue, visual language navigation, and task planning. Specifically, we provide a complete set of input and output interfaces for embodied agents' access, enabling agents to easily read task requirements and current environmental observations and then provide feedback results and obtain performance evaluations. Based on this platform, we evaluate popular multi-modal large language models and confirm the platform's values in evaluating embodied intelligence of different dimensions and difficulties. The contribution of this work can be summarized as follows.

- To the best of our knowledge, we take a pioneering step to construct a benchmark platform for embodied intelligence in an urban environment based on a real-world city.
- Based on the simulator, we set up a systematic benchmark including various and important tasks for embodied agents. The tasks reflect the multi-level and multi-dimensional capacities of embodied

Table 1: Comparisons of our EmbodiedCity with other related platforms.

| | Simulator | Engine | Environment | Visual | Agent | Sensing | Motion | Dataset | Task |
|---|---|---|---|---|---|---|---|---|---|
| CARLA (Dosovitskiy et al., 2017) | ✓ | UE | Fictional | ★★ | Vehicle | RGBD/GPS/Pose | Continous | ✓ | Autonomous Driving |
| MetaDrive (Li et al., 2022) | ✓ | Panda3D | Fictional | ★★ | Vehicle | RGBD/Lidar/Pose | Continues | ✓ | Autonomous Driving |
| nuScenes (Caesar et al., 2020) | ✗ | None | Real | ★★★ | Vehicle | RGB/Rader/Lidar | - | ✓ | Autonomous Driving |
| MetaUrban (Wu et al., 2024) | ✓ | PyBullet | Fictional | ★ | Robots | RGBD/Lidar | Continous | ✓ | Navigation |
| GRUtopia (Wang et al., 2024) | ✓ | Isaac Sim | Fictional | ★★ | Robots | RGBD | Continous | ✓ | Indoor |
| AerialVLN (Liu et al., 2023) | ✓ | UE | Fictional | ★★★ | Drone | RGBD | Continous | ✓ | VLN |
| CityNav (Lee et al., 2024) | ✓ | WebGL | Real | ★ | Drone | RGBD/GPS/Pose | Continous | ✓ | VLN |
| V-IRL (Yang et al., 2024) | ✓ | None | Real | ★★★ | - | RGB | Discrete | ✗ | Navigation/QA/Planning |
| TOUCHDOWN (Chen et al., 2019) | ✓ | None | Real | ★★★ | - | RGB | Discrete | ✓ | Navigation |
| AVDN (Fan et al., 2022b) | ✗ | None | Real | ★ | Drone | RGB | Discrete | ✓ | Navigation |
| **EmbodiedCity** | ✓ | UE | Real | ★★★ | All | All | Continous | ✓ | Scene Understanding /QA Dialogue/Navigation/Planning |

agents in the open outdoor urban environment. The ground truth data are carefully obtained with plenty of human efforts in data labeling.

- For the benchmark platform, we build the interface for embodied agents to observe, take action, and receive feedback. We further conduct evaluations on those popular large language models to verify the usability of our benchmark and have a quick look at the embodied intelligence level of these large language models. We released the executable program for this platform, and we have also designed a complete set of easy-to-use Python libraries and development documents[1].

## 2 RELATED WORK

**Autonomous Driving.** One of the related research directions is autonomous driving, in which the simulation platform can be used to train the driving and controlling algorithms for autonomous vehicles. CARLA (Dosovitskiy et al., 2017) constructs an environment for autonomous driving, which features urban streets and vehicle models equipped with sensing and control modules. However, it focuses on modeling road surfaces and traffic in small-town settings, with less emphasis on the realism of street layouts, urban planning, and building structures. MetaDrive Li et al. (2022) balances visual quality and efficiency by using Panda3D and Bullet to offer a lightweight driving simulator that supports research on generalizable reinforcement learning algorithms for vehicles. NuScenes (Caesar et al., 2020) provides a perception dataset captured on real roads using cameras, radars, and lidar. However, these works are designed as simulators and datasets for autonomous driving and are less suitable for broader urban embodied task research.

**Embodied-AI Platforms based on Real City.** CityNav (Lee et al., 2024) has developed a WebGL-based simulator grounded in real cities, but the tasks supported by this platform are quite limited, and actions in the navigation task are discrete, with a large gap to the real-world navigation. Both V-IRL (Yang et al., 2024) and TOUCHDOWN (Chen et al., 2019) are built on Google Maps, offering the highest visual realism, yet they only support discrete movements. AVDN (Fan et al., 2022b) provides agents with RGB sensing inputs derived from satellite images, which are of low precision. That is, these platforms based on real cities struggle to meet the high-quality, high-precision, and continuous sensing and movement requirements of embodied agents.

**Embodied-AI Platforms based on Fictional City.** MetaUrban (Wu et al., 2024) has developed a streetscape generation simulator that facilitates top-down design for road layout, object placement, and dynamic urban traffic generation, providing convenience for navigation simulation of ground-based robots. However, it focuses solely on the neighborhood level, neglecting city-scale design, thereby limiting the activity space of agents. Additionally, object modeling is relatively coarse, at a sticker level, which may pose challenges for applications like image recognition algorithms. GRUtopia (Wang et al., 2024) offers a variety of navigation, dialogue, and manipulation tasks for different types of ground robots within multiple indoor scenarios such as hospitals, restaurants, and libraries (it is still an indoor environment). AerialVLN (Liu et al., 2023) primarily addresses the vision-and-language navigation problem for drones in urban environments. However, these benchmarks are primarily constructed within fictional cityscapes and are simplified compared to real

---

[1]All of the software, Python library, codes, datasets, tutorials, and real-time online service are available on this anonymous website: `https://embodied-ai.city`.

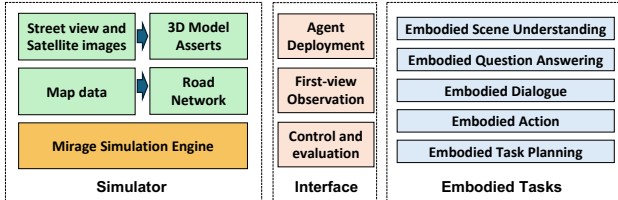

Figure 2: The framework of our platform, including simulator, interface, and embodied tasks.

urban scenarios. Unlike them, our platform is built with multiple types of city elements based on the real-world city with various agents and systematic EmbodiedAI tasks.

Specifically, we propose an urban embodied benchmark that creates a highly realistic large-scale city model using Unreal Engine. It ensures high fidelity in simulation and supports various types of sensing and control for both drones and ground robots, offering datasets for five types of embodied tasks. We compare these representative ones with the proposed EmbodiedCity benchmark, as listed in Table 1. We can observe that EmbodiedCity is the first platform with a high-quality 3D real environment based on a real city, supporting various agents, continuous decision-making, and systematic benchmark tasks for embodied intelligence.

## 3 THE BENCHMARK PLATFORM

The core of this benchmark platform is an environment (as shown in Figure 1). Based on this environment, we have established interfaces for agents to be deployed in the environment, read input with first-view observations, and make decisions. The overall workflow is shown in Figure 2. In this section, we will elaborate on the detailed information, the 3D environment, the interface for embodied agents, SDK, and online access.

### 3.1 3D ENVIRONMENT

The basic environment of the simulator includes a 2.8km × 2.4km district in Beijing, one of the biggest cities in China, where we meticulously build 3D models for buildings, streets, and other outdoor elements, ensuring high-fidelity urban simulations, all hosted by Unreal Engine 5.3[2]. In addition to this business district, we have also developed a nearby residential area that features detailed interior modeling, allowing the simulator to support both indoor and outdoor tasks. Below are the key components that make up the environment:

- Buildings. We first manually use Blender[3] to create the 3D assets of the buildings, for which we use the streetview services of Baidu Map[4] and Amap[5]. The city-level detail includes approximately 200 buildings, encompassing a variety of types such as office towers, shopping malls, residential complexes, and public facilities. These models are textured and detailed to closely resemble their real-world counterparts to enhance realism in the simulation.
- Streets. The city contains a total of 100 streets, with a combined length of approximately 50 km. The streets are modeled to include all necessary components such as lanes, intersections, traffic signals, and road markings. We also incorporate pedestrian pathways, cycling lanes, and parking areas. Data from traffic monitoring systems and mapping services help ensure that the street layout and traffic flow patterns are accurate and realistic.
- Vehicles and Pedestrians. Dynamic elements such as vehicles and pedestrians are simulated to move realistically within the environment. The simulation algorithms for these elements are based on the Mirage Simulation System (Zhang et al., 2022), providing realistic interactions and behaviors that mimic real-world traffic and pedestrian dynamics.
- Other Elements. Besides streets and buildings, other elements include street furniture (benches, streetlights, signs), vegetation (trees, shrubs, lawns), and urban amenities (bus stops, metro entrances, public restrooms). Over 6k urban elements are created using Blender, based on real-world

---

[2]https://www.unrealengine.com/
[3]https://www.blender.org/
[4]https://map.baidu.com/
[5]https://amap.com/

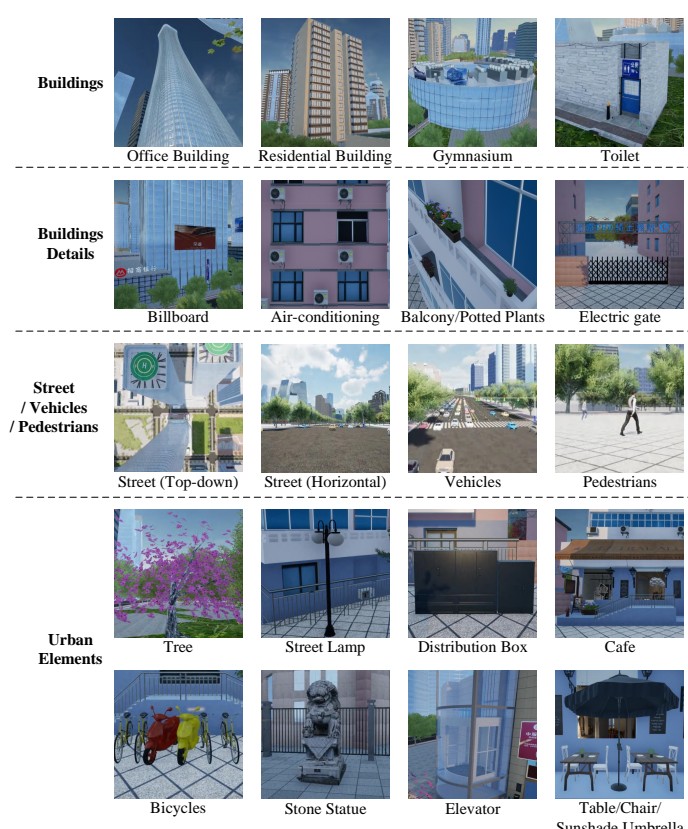

Figure 3: The image showcases various components of our city simulator.

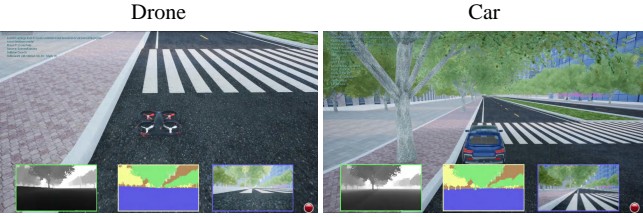

Figure 4: Integration of agent interface within the Unreal Engine city simulator environment, showcasing the simulation of both drones and unmanned vehicles (cars).

references from the streetview services mentioned above, further enhancing the authenticity of the urban simulation.

We further provide detailed information on the environment, as shown in Figure 3. Our city simulator is a tool designed for urban planning, analysis, and autonomous vehicle simulation. It offers superior capabilities compared to other available simulators, featuring high-resolution 3D models and real-time data integration for an exceptionally realistic and dynamic representation of urban environments. The simulator's customization options allow users to model diverse scenarios and explore various urban elements, from detailed building features to specific street-level details. Specifically, it supports simulations for drones and unmanned vehicles, making it an invaluable resource for testing and optimizing autonomous sensing, navigation, and planning in urban settings.

## 3.2 INTERFACE OF EMBODIED AGENTS

With the environment of Unreal Engine, we further build the interface of embodied agents to ensure the agents can indeed embody themselves in the system. To implement it, we develop the input/output

interfaces based on AirSim[6], based on which the observations can be conducted in a first-view manner, and the control actions include motion, velocity, accelerated velocity, etc. This provides a robust framework for simulating realistic interactions and behaviors for both drones and vehicles.

- Observations. The observations for the embodied agents are designed to replicate the sensory inputs available to real-world agents. For drones, these include image data such as RGB images (color images from the camera), depth images (showing the distance of each pixel from the camera), and segmentation images (semantic segmentation for different objects in the scene). Sensor data like IMU data (accelerometer and gyroscope data for measuring acceleration and angular velocity), GPS data (providing global positioning coordinates), and LiDAR data (3D point cloud information of the environment) are also included. State information such as position (current coordinates x, y, z), speed (current velocity vx, vy, vz), and attitude (current orientation roll, pitch, yaw) is vital. For vehicles, the observations include similar image data and sensor data. Additionally, state information for vehicles includes the position, speed, attitude, and wheel angle (current wheel steering angle).
- Actions. The actions of the embodied agents mimic realistic controls similar to those used by air drones and vehicles. For drones, motion control involves setting target positions (x, y, z), target velocities (vx, vy, vz), and target orientations (roll, pitch, yaw). Camera control allows for viewpoint adjustments (changing camera direction pan, tilt), and other controls include starting or stopping the drone's flight. For vehicles, driving control includes steering angle (setting the steering wheel angle), acceleration (setting throttle), braking (setting brake force), and gear shifting (switching between gears: forward, reverse, neutral). Camera control for vehicles also allows for viewpoint adjustments, and other controls include starting or stopping the vehicle's movement.

## 3.3 SDK AND ONLINE ACCESS

To make the simulator less difficult to use, we developed a Python client software development kit (SDK) and a Python proxy server based on the HTTP protocol on top of the AirSim interface. The Python proxy server is used to convert client requests into AirSim interface calls and return AirSim responses. With this Python proxy server, we shield client development from the outdated, non-standard event loop asynchronous module employed by AirSim, and allow for easier remote access using a variety of HTTP-based infrastructure. The HTTP protocol they use mainly adopts JSON format for data transfer and sends images, for which the details are available in our open-source code repository. This Python client SDK implements synchronous and asynchronous methods based on the standard async mechanism to support users writing highly concurrent programs, such as concurrent requests for large models with the simulator.

Based on the above open transport protocol and Python client SDK, we build an online platform for users to try out. The online platform supports the simultaneous simulation and control of up to 8 agents. Users can acquire control of one or more idle agents and manipulate their movements via keyboard keys, the web GUI, or even an online Python code editor. Users can also watch a first-person view of all the agents via live video streaming. The platform is open for registration and use since we hope that the open online platform will inspire more ideas and explorations, such as collaborations on embodied agents in urban environments.

## 4 BENCHMARK TASKS ON EMBODIED AI IN OPEN CITY ENVIRONMENTS

Using the constructed simulator platform, we have built a dataset comprising 87.1k cases, as shown in Table 2. This dataset includes five essential embodied tasks, which cover various aspects of embodied intelligence abilities. Specifically, the intelligent agents in the open world are expected to have three kinds of human-like abilities: perception, reasoning, and decision-making. For perception, we consider the task of embodied first-view scene understanding; for reasoning, we consider the task of embodied question answering and dialogue; for decision-making, we consider the task of embodied action (visual-language navigation) and embodied task planning. For a better understanding, we present five tasks in Figure 5.

- **Embodied first-view scene understanding**. The first-view scene understanding requires the agent to comprehend its surrounding environment and give an accurate description, which could consid-

---

[6]https://github.com/microsoft/AirSim

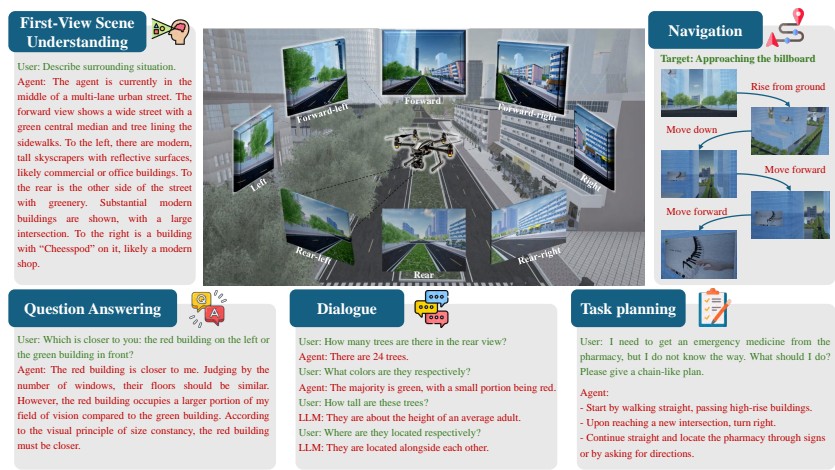

Figure 5: Illustration of the embodied tasks in the urban environment.

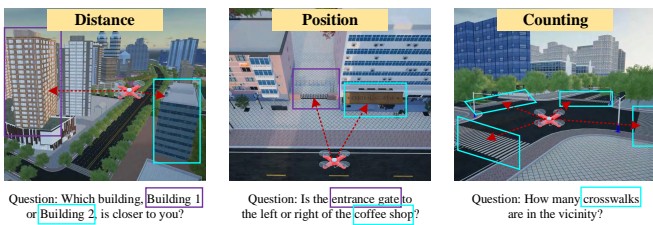

Figure 6: Task examples. Three QA benchmarks are established for evaluating embodied agents: Distance, Position, and Counting. Agent and objects in questions are depicted in the figure.

ered a basic ability for further tasks. In our benchmark, we observe from different perspectives at the same location, generating a set of RGB images, *i.e.*, the input of scene understanding, and the output is the textual description for the given scene images.

- **Embodied question answering**. The embodied agent can be further queried in natural language about the environment. We have designed three types of embodied question-answering tasks: distance, position, and counting, as shown in Fig. 6. **Distance** questions involve determining the relative distance between the agent and surrounding city elements, such as "Which is closer to me, the blue building or the red building?" and "Approximately how many meters away is the building ahead of me?". **Position** questions assess the understanding of spatial relationships in the environment, such as "Is object A to the left or right of object B?" and "Which street is in front, Street A or Street B?". **Counting** questions evaluate the agent's accurate perception of the environment, such as "How many crosswalks can be seen in a full circle?". Therefore, the input includes both the first-view RGB images and a query about the environment. The output should be the direct textual responses to the question.

- **Embodied dialogue**. Despite the task of embodied question answering, a more complex embodied task close to the reasoning ability is embodied dialogue. Specifically, embodied dialogue involves ongoing interactions where the agent engages in a back-and-forth conversation with the user. This requires maintaining context and understanding the flow of dialogue. Therefore, the input includes embodied observations and multi-round queries, and the output is the multi-round responses.

- **Embodied action (navigation)**. Embodied Action, often referred to as Vision-and-Language Navigation (VLN), focuses on enabling an agent to navigate an environment based on natural language instructions. The input combines visual perception and natural language instructions to guide the agent through complex environments, and the output consists of the action sequences following the language instructions.

- **Embodied task planning**. Most of the time, decision-making in the real world does not have explicit instructions; otherwise, there is only an unclear task goal. Thus, it is significant for the embodied agents to be able to compose the complex and long-term task goals into several sub-tasks, which we refer to as embodied task planning. The input is the first-view observations and a given

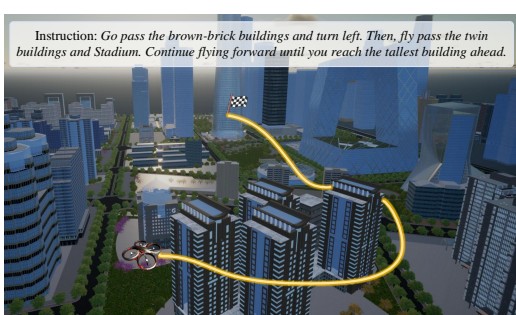

Figure 7: Illustration of the embodied VLN tasks.

Table 2: **Datasets statistics.** The dataset includes details on how it was collected, how the ground truth was obtained, the number of cases, and the token count of the dataset's text portion.

| Task | 2D input | 3D input | # words(avg.) | Prompt Collection | Ground Truth | # Cases |
|---|---|---|---|---|---|---|
| Embodied scene understanding | ✓ | ✓ | 57.3 | Human | AutoGen+Refinement | 12.2k |
| Embodied QA | ✓ | ✓ | 16.5 | AutoGen+Human | AutoGen+Refinement | 50.4k |
| Embodied dialogue | ✓ | ✓ | 48.7 | AutoGen+Human | AutoGen+Refinement | 12.6k |
| Embodied VLN | ✓ | ✓ | 56.5 | Human | Human | 1.3k |
| Embodied task planning | ✗ | ✗ | 66.0 | AutoGen+Human | AutoGen+Refinement | 10.6k |

Table 3: Results of embodied first-view scene understanding.

| Model | BLEU-1 | BLEU-2 | BLEU-3 | BLEU-4 | ROUGE | METEOR | CIDEr |
|---|---|---|---|---|---|---|---|
| Fuyu-8B | 40.25 | 20.26 | 8.40 | 1.57 | 17.29 | 15.80 | 21.55 |
| Qwen-VL | 40.57 | 17.59 | 5.90 | 0.98 | 14.61 | 19.13 | 18.40 |
| Claude 3 | 57.38 | 31.73 | 16.83 | 7.19 | 21.60 | 29.00 | 29.20 |
| GPT-4 Turbo | 54.01 | 27.63 | 12.73 | 4.53 | 21.99 | 28.48 | 22.39 |

natural language described task goal, and the output should be a series of sub-tasks that the agent plans to execute.

**Data labeling and human refinement** For different types of tasks, we employed various methods to expand and annotate the data, as shown in the Table 2. More details about collecting the labels can be found in the supplemental material. During the construction process, human refinement plays an important role[7]. We spend a lot of human effort obtaining the ground-truth data for these tasks.

## 5 EVALUATION OF LARGE LANGUAGE MODELS

We select popular and representative multi-modal large language models for evaluation to verify the application value of our benchmark and test their abilities for embodied tasks in the urban environment. The considered models include Fuyu-8b, Qwen-VL, Claude 3, GPT-4 Turbo.

**Task I: Embodied first-view scene understanding.** The results of the performance evaluation are presented in Table 3, with typical evaluation metrics: BLEU (Papineni et al., 2002), ROUGE (Lin, 2004), METEOR (Banerjee & Lavie, 2005), and CIDEr (Vedantam et al., 2015). From the results we have the following observations:

- Claude 3 has shown the best performance on the task of embodied scene understanding, with the best performance on almost all metrics. Actually, in this task, the different metrics have similar distinguishing abilities, *i.e.,* a more with better performance on one metric is likely to be better on another metric.
- Larger scale models steadily outperform those smaller ones. As we can observe, Fuyu-8B and Qwen-VL have similar parameter sizes (7B-8B), which are far smaller than Claude 3 and GPT-4 Turbo.

**Task II: Embodied question answering**

---

[7]Illustrated in Figure 9 in the Appendix.

Table 4: Results of embodied question answering.

| Type | Model | BLEU-1 | BLEU-2 | BLEU-3 | BLEU-4 | ROUGE | METEOR | CIDEr | Sentence-BERT |
|---|---|---|---|---|---|---|---|---|---|
| Distance | Fuyu-8B | 20.19 | 18.36 | 16.39 | 14.64 | 31.55 | 20.34 | 22.56 | 45.13 |
| | Qwen-VL | 55.77 | 48.43 | 40.90 | 31.94 | 65.33 | 61.73 | 33.30 | 47.12 |
| | Claude 3 | 49.34 | 41.88 | 34.10 | 23.44 | 60.51 | 55.29 | 29.84 | 39.50 |
| | GPT-4 Turbo | 76.63 | 72.17 | 68.57 | 65.51 | 80.16 | 77.10 | 61.44 | 63.92 |
| Position | Fuyu-8B | 7.46 | 0.15 | 0 | 0 | 18.94 | 4.40 | 12.86 | 41.35 |
| | Qwen-VL | 7.88 | 4.63 | 3.81 | 0.83 | 18.03 | 22.00 | 16.62 | 19.33 |
| | Claude 3 | 7.57 | 5.85 | 4.37 | 1.56 | 19.04 | 34.28 | 18.82 | 20.46 |
| | GPT-4 Turbo | 64.54 | 61.85 | 59.44 | 55.31 | 70.72 | 68.87 | 58.45 | 69.33 |
| Counting | Fuyu-8B | 12.00 | 7.15 | 1.07 | 0.40 | 16.45 | 15.41 | 8.87 | 38.94 |
| | Qwen-VL | 5.49 | 1.19 | 0.10 | 0 | 11.46 | 17.89 | 3.58 | 41.29 |
| | Claude 3 | 6.08 | 4.33 | 2.79 | 2.13 | 10.54 | 16.82 | 7.95 | 33.87 |
| | GPT-4 Turbo | 12.84 | 8.81 | 4.33 | 2.78 | 19.26 | 20.18 | 11.56 | 41.07 |

Table 5: Results of embodied dialogue.

| Model | BLEU-1 | BLEU-2 | BLEU-3 | BLEU-4 | ROUGE | METEOR | CIDEr | Sentence-BERT |
|---|---|---|---|---|---|---|---|---|
| Fuyu-8B | 29.05 | 16.73 | 8.24 | 4.30 | 28.53 | 30.12 | 14.47 | 55.57 |
| Qwen-VL | 17.91 | 9.54 | 3.90 | 2.03 | 19.33 | 19.65 | 10.30 | 52.10 |
| Claude 3 | 24.86 | 18.02 | 13.14 | 9.70 | 29.06 | 38.56 | 28.62 | 73.81 |
| GPT-4 Turbo | 41.77 | 34.27 | 27.82 | 23.26 | 42.29 | 51.72 | 35.64 | 76.62 |

The results of the performance evaluation are presented in Table 4. To enhance the semantic evaluation of QA tasks, sentence-BERT (Reimers, 2019) metric was incorporated. From the results, we have the following observations:

- GPT-4 Turbo achieves a very significant performance improvement against all other models, of which the average improvement is larger than 100%. This may be explained by the GPT-4's stronger ability to handle textual data.
- Smaller models are very unsteady on three types of tasks, counting, property, and position, for which some metrics are 0.

**Task III: Embodied dialogue** The results of the performance evaluation are presented in Table 5, from which we have the following observations:

- GPT-4 Turbo shows the best performance with significant gain, which could be explained by the long-context abilities, which is the major requirement of multi-round conversions.
- The poor performance of Qwen-VL provides insights that it may be a promising solution to combine large language models that do not support vision input, such as QWen.

**Task IV: Embodied VLN** The results of the performance evaluation are presented in Table 6, from which we have the following observations:

- Both GPT-4o and Claude 3 achieve the best performance on SR and SPL. And GPT-4o also has the lowest NE compared to other models. This implies that GPT-4o has the strongest spatial reasoning capacity that always navigates the drone in the correct direction.
- Chinese LLM (Qwen-VL) has a significant performance drop against English LLM. Qwen-VL is 12% and 15% lower than the best-performing model in SR and SPL metrics, respectively. This result can be attributed to the superior performance of the English LLM in understanding English task descriptions and applying them to action reasoning.
- All models perform better on short navigation tasks than long navigation tasks which involve longer reasoning chains and more dramatic scene changes, causing higher failure rates.

**Task V: Embodied task planning** The results of the performance evaluation are presented in Table 7, from which we have the following observations:

- Claude-3 achieves the best performance on embodied task planning. Actually, task planning relies more on decision-making ability with common sense and contextual information. Therefore, it pay less attention to the multi-modal understanding ability.
- Smaller LLMs show poorer performance, but the performance gap is acceptable, which inspires us to deploy mixture-architecture agents, combining the strengths of larger and smaller LLMs.

Table 6: Results of embodied vision-and-navigation.

| Model | Short | | | Long | | | Mean | | |
|---|---|---|---|---|---|---|---|---|---|
| | SR/% | SPL/% | NE/m | SR/% | SPL/% | NE/m | SR/% | SPL/% | NE/m |
| Qwen-VL | 33.33 | 29.60 | 67.30 | 8.33 | 6.67 | 145.3 | 22.22 | 19.33 | 120.44 |
| Claude 3 | 76.92 | 75.60 | 139.11 | 20.00 | 19.65 | 185.48 | 34.90 | 34.25 | 162.35 |
| GPT-4 Turbo | 60.90 | 55.21 | 95.93 | 15.62 | 14.16 | 127.87 | 27.71 | 25.12 | 111.92 |
| GPT-4O | 76.92 | 75.60 | 77.23 | 20.00 | 19.65 | 102.98 | 34.90 | 34.25 | 90.11 |

Table 7: Results of embodied task planning.

| Model | BLEU-1 | BLEU-2 | BLEU-3 | BLEU-4 | ROUGE | METEOR | CIDEr |
|---|---|---|---|---|---|---|---|
| Fuyu-8B | 15.11 | 6.37 | 1.71 | 0.45 | 14.72 | 19.11 | 16.84 |
| Qwen-VL | 20.28 | 9.10 | 3.75 | 1.44 | 19.42 | 17.90 | 11.36 |
| Claude 3 | 29.21 | 16.22 | 9.17 | 4.40 | 22.85 | 31.58 | 21.78 |
| GPT-4 Turbo | 28.23 | 13.72 | 6.26 | 2.82 | 21.61 | 28.47 | 16.41 |

## 6    DISCUSSIONS AND LIMITATIONS OF THE BENCHMARK

**Application and promotion on EmbodiedAI.** The benchmark not only serves as the pure evaluation of the large language model or LLM agents but also could be a sim2real tool that supports the pre-training or pre-testing before being deployed to the real-world city environment. For the types of agents, the benchmark does not set constraints. That is, the agent deployed could be a robot or air drone. For example, the input of a robot may only include the RGB images, and for air drones, the input can also contain the radar signals. The degree of freedom of different agents could also be different. This platform expands the boundaries of embodied functions, promotes the category of embodied intelligence, and can effectively support the further development of this field.

**Human refinement.** When constructing the benchmark, we put a lot of effort into using human refinement steps to filter out low-quality responses or revise incorrect answers provided by LLMs. It is worth noticing that the paradigm of combining large language models and human crafts has recently become widely used since large language models accurately and skillfully generate various responses (but may be totally wrong). The key challenge here is accuracy rather than diversity, and thus, the human efforts to refine the answers are quite essential and useful. On the other hand, the cost of collecting all the responses with pure human labor is not affordable. Therefore, using the large language models does not introduce a large bias.

**Extensions of benchmark.** In our constructed benchmark, we consider five types of embodied tasks, scene description, embodied question answering, embodied dialogue, visual-language navigation, and embodied task planning. From a perspective of human-like critical abilities, these tasks well cover the three most significant aspects: perception, reasoning, and decision-making. The follow-up work, based on the simulation environment, promises to extend to more tasks, of which the potential tasks could be as follows. (1) Multi-agent Collaboration: Introducing tasks that require coordination and communication between multiple agents to achieve common goals. (2) Human-Agent Interaction: Creating scenarios where human users interact with agents necessitates a more sophisticated understanding of human behavior and natural language. (3) Adaptability and Learning: Implementing tasks that test an agent's ability to learn from its environment and adapt to new and unforeseen scenarios.

## 7    CONCLUSION AND FUTURE WORK

In this work, we take a pioneering step by building a systematic benchmark for embodied intelligence in an open city environment. The benchmark contains a 3D city simulator, easy-to-use interfaces for embodied agents, and five kinds of embodied tasks. We further evaluate the intelligence capacities of some popular multi-modal large language models, which verify the rationality of the constructed benchmark. We also plan to evaluate large language model agents' performance and embodied intelligence level in the real city environment via a Sim2Real paradigm, which can further validate the application value of the benchmark. We believe this work can help narrow the gap between existing embodied intelligence research and the ultimate goal of artificial general intelligence.

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

# A SUPPLEMENTARY MATERIALS

## A.1 SIMULATOR

In our city simulator, AirSim serves as a powerful plugin to facilitate realistic simulations of drones and unmanned vehicles. These autonomous systems leverage AirSim's robust observation and action mechanisms to navigate and interact with the urban environment.

For drones, the observation process involves capturing high-resolution images and sensor data from multiple perspectives, including RGB, depth, and segmentation views. These observations enable the drone to perceive its surroundings accurately, identify obstacles, and navigate complex urban landscapes. The action space for drones includes vertical movements (up and down), horizontal movements (forward, backward, left, and right), and rotational adjustments (yaw, pitch, and roll). This comprehensive action space allows drones to maneuver precisely and efficiently in three-dimensional urban environments.

Similarly, for unmanned vehicles, observation is achieved through an array of sensors that provide comprehensive environmental data, including visual feeds and depth information. This allows the vehicle to detect road features, other vehicles, pedestrians, and potential hazards. The action space for unmanned vehicles includes steering (left and right), acceleration (forward movement), and braking (deceleration and stopping). These actions ensure that the vehicle can navigate urban streets safely and efficiently by making real-time adjustments based on its observations.

By integrating AirSim into our city simulator, we provide a detailed and realistic platform for testing and optimizing the performance of autonomous drones and vehicles in urban settings.

## A.2 OPEN INTERFACE

Our city simulator feature an open API interface. This API will provide users with the ability to programmatically access and manipulate various aspects of the simulator. Through this interface, users can control camera perspectives, navigate virtual characters, retrieve environmental data, and perform other interactive tasks. The API will be designed with robust measures to ensure safe and authorized access, thereby making our simulator a versatile tool for both research and practical applications.

## A.3 BENCHMARK AND DATASET

- **Embodied first-view scene understanding.** We randomly walk around the city and record the surrounding RGB observations upon reaching a location. For each case, the prompts are fixed and can therefore be designed manually. For the ground truth, we first generate embodied descriptions using the VLM. Then we manually review and correct each response, as shown in Table 8. The refinement process involves five categories of raw responses:

  1. Object Counting: The question involves counting a specified object.
  2. Object Existence: The response asserts the presence of objects, which may or may not actually exist.
  3. Object Position: The response describes the spatial relationship between buildings or objects.
  4. Negative Response: Indicates that the question cannot be answered and will be discarded.
  5. Unnecessary Content: The response includes redundant information that could impact the calculation of evaluation metrics.

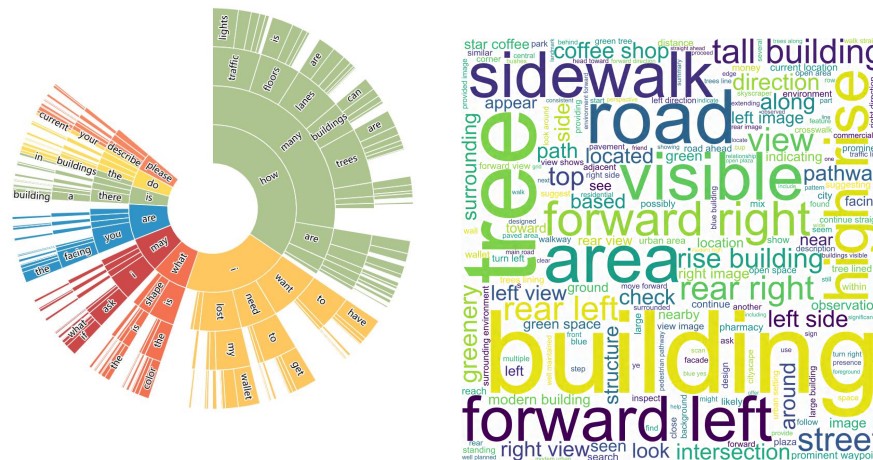

(a) The keywords of questions in the task of embodied question answering. (b) The word cloud of the task of embodied question answering.

Figure 8: Illustration of the involved topics and keywords in the task of embodied question answering in our benchmark.

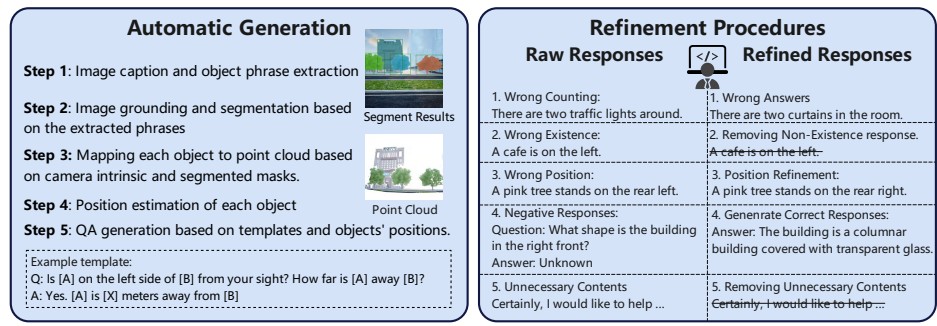

Figure 9: The refinement procedures when constructing the benchmark, which aims to address the errors in raw responses, inspired by Huang et al. (2024).

- **Embodied question answering.** Similarly, upon randomly arriving at a certain location, we record the surrounding RGB observations and specifically inquire about details of the current situation, such as the color of buildings in a particular direction or the number of trees nearby. To generate questions with urban characteristics, we have GPT-4o select questions that match the current scene based on the aforementioned images, in conjunction with a pre-generated question bank created manually. The refinement examples are listed in Table 9.

- **Embodied dialogue.** This task is an enhanced version of question answering, requiring continuous question and dialogue responses. It further tests the logical reasoning and vision-language comprehension capabilities of large models. The processes for prompt collection and ground truth acquisition are similar. Examples of dialogue refinement are shown in Table 10

- **Embodied VLN.** In navigation tasks, it is crucial to reasonably select the agent's starting and target points within the city simulator. The navigation difficulty increases with the distance between the starting point and the target point. Additionally, the target point must be distinctive to ensure the uniqueness of the spatial location referred to by the textual description. The process of Vision-and-Language Navigation (VLN) is dynamic, requiring continuous interaction with the simulator. Each decision at every step influences the subsequent observation, thereby affecting the next decision. Consequently, the input and ground truth for each case are obtained through human annotation. The input consists of the agent's starting coordinates and textual instructions, while the ground truth comprises the route trajectory and the target coordinates.

Table 8: Examples of first-view scene understanding refinement.

| Types | Raw Responses | Refined Responses |
|---|---|---|
| Wrong Counting | In front of us, there's a large building There seem to be three such buildings visible within this frame. | There are several tall buildings made up of glass windows. The surroundings include several large, architecturally modern buildings. |
| Wrong Existence | Multiple cars parked along the roadside, with varying sizes indicating depth perception Above them, the sky appears clear blue with white clouds scattered throughout. | *The unnecessary contents will be removed.* |
| Wrong Position | The scene shows an urban road perspective view in daylight conditions. On both sides of the road stand two-story high walls made of dark-colored stone blocks. | You are in a cityscape with modern and tall buildings. he view shows a tall, modern building made of concrete or stone on the right. |
| Negative Responses | As an AI language model, I do not have physical senses or locations in the real world. The user is currently standing in an urban area at night time. | Based on the observations from the eight directions, it seems you are in an urban environment surrounded by tall modern buildings, likely in a city center. The user is currently in an urban area at daytime, standing near a road intersection. |
| Unnecessary Contents | The scene shows an urban street viewed from above at an angle of approximately 45 degrees. The scene shows an urban street viewed from above at a slight angle. | *The unnecessary contents will be removed.* |

Table 9: Examples of question answering refinement.

| Types | Raw Responses | Refined Responses |
|---|---|---|
| Wrong Counting | Q: How many traffic lights can be observed around in total? A: None. | Q: How many traffic lights can be observed around in total? A: 1 traffic light can be observed. |
| Wrong Existence | Q: Is there a building on the left side? A: There is no building visible in any of the provided inputs. | Q: Is there a building on the left side? A: Yes, there is a building on the left side. |
| Wrong Position | Q: Are you facing the road, the building, or the greenery? A: Road. | Q: Are you facing the road, the building, or the greenery? A: You are facing the greenery. |
| Negative Responses | Q: How many lanes are there on the road behind? A: There are no lanes on the road behind. | Q: How many lanes are there on the road behind? A: Two lanes. |
| Unnecessary Contents | Q: How many floors is the tallest building around? A: The spiral shaped skyscraper is clearly the tallest structure in the cityscape. | *The unnecessary contents will be removed.* |

- **Embodied task planning.** Compared to the question-answering task, this task presents an increased difficulty level. It requires the output of a step-by-step plan to achieve a complex objective. The processes of prompt collection and ground truth acquisition are similar to those in question answering. The refinement types consist of abnormal logic, negative responses, non-planned, and unnecessary contents, as shown in Table 11.

## A.4 EXPERIMENTAL SETUP

To evaluate the performance of embodied first-view scene understanding, question answering, dialogue, and task planning, we use the following metrics: BLEU-1, BLEU-2, BLEU-3, BLEU-4, ROUGE, METEOR, and CIDEr. BLEU (Papineni et al., 2002) scores measure n-gram precision at

Table 10: Examples of dialogue refinement.

| Types | Raw Responses | Refined Responses |
|---|---|---|
| Wrong Counting | There are 3 trees beside the road. I see a tall building straight ahead. | There are four trees at the junction. I see three high ups in front of me. |
| Wrong Existence | Yes, there is a crosswalk near the road. Yes, there is an intersection ahead which may require turning or stopping for traffic lights. | There are no prominent waypoints. No, there are no prominent waypoints visible in the observations. |
| Wrong Position | The building is situated next to the road in front of the viewer. It's situated beside the road in front of me. | The building on the left is located on the left side of the road ahead. Adjacent to the road ahead on the left side. |
| Negative Responses | No, there is no mention of waypoints in the given information about the environment. The locations cannot be determined based solely on the provided information. | Yes, there is a prominent intersection with traffic lights, which can be considered a waypoint. All images depict various angles of one major intersection with traffic lights. |
| Unnecessary Contents | Certainly, l would like to help ... I'm happy to judge the scene for you, and my verdict is.... | *The unnecessary contents will be removed.* |

different lengths (1 to 4). ROUGE (Lin, 2004) focuses on recall, capturing how much of the reference text is covered by the generated text. METEOR (Banerjee & Lavie, 2005) combines precision and recall while accounting for synonyms and word forms. CIDEr (Vedantam et al., 2015) evaluates the similarity of generated descriptions to reference descriptions, particularly in image description tasks, using TF-IDF weighting. Sentence-BERT (Reimers, 2019) is a modification of the BERT network designed to derive semantically meaningful sentence embeddings for efficiently comparing and measuring the similarity between sentences. These metrics collectively provide a comprehensive assessment of model performance.

## A.5 BASELINES

We utilize powerful large models in multimodal AI, each bringing unique strengths and capabilities to the embodied tasks.

**Fuyu-8B** (AdeptAI, 2024), developed by Adept AI, is a multimodal model designed to assist knowledge workers. Its strengths include a simple architecture and training process that facilitate understanding, scalability, and deployment, as well as the ability to handle arbitrary image resolutions, charts, and screen images with fine-grained localization. It offers fast response times (under 100 milliseconds for large images) and performs well on standard image understanding benchmarks such as visual question answering and natural image captions. This model is used to evaluate the capabilities of open-source multimodal large models.

**Qwen-VL** (Bai et al., 2023), proposed by Alibaba Cloud, is a large-scale vision-language model that supports image, text, and detection boxes as input, and produces text and detection boxes as output. It excels in multilingual dialogue, interleaved multi-image dialogue, Chinese open-domain localization, and fine-grained image recognition.

**Claude 3** (ClaudeTeam, 2024), introduced by Anthropic, is a large-scale language model designed for enterprise use, balancing speed and performance. It offers three levels—Haiku, Sonnet, and Opus—catering to different tasks and is known for being secure, reliable, and customizable. This paper primarily utilizes Claude 3 Haiku.

**GPT-4 Turbo** (OpenAI, 2024), released by OpenAI, is a highly intelligent model that supports both image and text inputs, generating text outputs. As one of the most powerful multimodal large models, its performance delineates the optimal embodied capability boundaries of current models.

Table 11: Examples of task planning refinement.

| Types | Raw Responses | Refined Responses |
|---|---|---|
| Abnormal logic | Q: I want to have a cup of coffee at ALL-Star coffee shop, but I have not brought any money. What should I do? Please give a chain-like plan.
A:
1. Look for any loose change or money on the ground or in the vicinity.
2. If you find some money, use it to purchase a coffee at the ALL-Star coffee shop.
3. If you don't find any money, look for any odd jobs or tasks you can perform around the area in exchange for a cup of coffee, such as helping someone carry their groceries or cleaning up litter. | *The abnormal logic will be removed.* |
| Negative Responses *(Wrong Position)* *(Wrong Existence)* | Q: I need to get an emergency medicine from the pharmacy, but I do not know the way. What should I do? Please give a chain-like plan.
A:
1. Go forward towards the blue building in the center of the image.
2. Turn left and follow the road.
3. Continue straight, crossing the pedestrian crossing.
4. Turn right and follow the road.
5. The pharmacy should be visible on your right.
6. Proceed to the pharmacy and obtain the emergency medicine. | *You cannot find a pharmacy along this step. The wrong plan will be removed.* |
| Non-planned | Q: I lost my wallet nearby, and now I need to find it. What should I do? Please give a chain-like plan.
A: A good way is that we can use the camera in reverse to see if our wallet appears in this picture. | *That's not the plan. It will be removed.* |
| Unnecessary Contents | A: Head forward along the road. There are traffic lights and crosswalks. | *The unnecessary contents will be removed.* |

## A.6 CASE STUDY

### A.6.1 TASK I: EMBODIED FIRST-VIEW SCENE UNDERSTANDING

As shown in Figure 10, fuyu-8B provides a detailed scene description, focusing on visual elements such as the floor, wall, bench, and potted plants. It mentions geometric shapes and artistic perspectives but does not explicitly state the general location or surrounding environment. The model's strengths lie in its detailed visual analysis and focus on specific elements. However, it lacks a holistic description of the location and environment, which limits its overall effectiveness in this task.

Qwen-VL emphasizes its inability to sense physical locations but infers that the scene could be an urban setting with buildings, roads, sidewalks, and possibly parks or plazas. Although it provides a general guess, it lacks specific details and a comprehensive scene description. The model's strengths include acknowledging its limitations and providing a broad inference, but its weaknesses are evident in the lack of detailed specifics and a thorough scene description.

Claude 3 identifies the scene as a large open plaza in an urban environment, surrounded by tall buildings and various types of trees. It notes the tiled ground, benches, and potential for a gathering space. The description is coherent and aligns well with the observed images. Claude 3's strengths are its comprehensive and coherent description, identification of key features, and correct inference of the environment.

GPT-4 Turbo describes the scene as a large urban plaza surrounded by high-rise buildings with a tiled ground and arranged trees. It mentions the presence of a pedestrian area and suggests the scene is a public space within a city, characterized by a blend of natural elements and urban architecture.

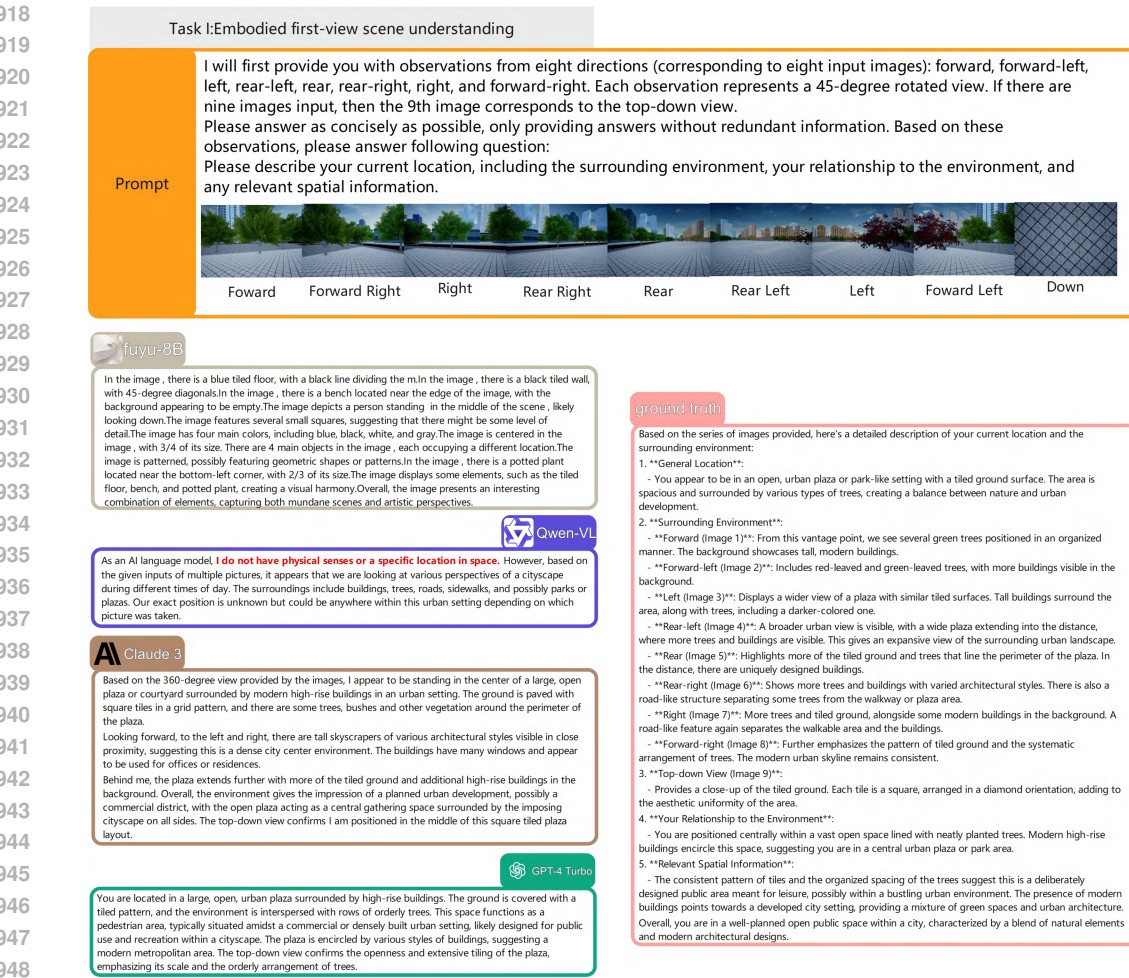

Figure 10: Embodied first-view scene understanding task involves describing one's current location, surrounding environment, relationship to the environment, and any relevant spatial information based on observations from eight directions (forward, forward-left, left, rear-left, rear, rear-right, right, and forward-right) and one top-down view image. The specific outputs of different methods are listed separately.

The model's strengths include its detailed and accurate description, along with information about the environment and its potential uses.

Claude 3 and GPT-4 Turbo excel in providing detailed, accurate, and coherent descriptions, closely aligning with the ground truth. Their responses demonstrate a strong understanding of the scene, balancing specific visual elements with broader contextual insights. Fuyu-8B and Qwen-VL offer valuable observations but fall short of delivering comprehensive descriptions. This analysis highlights the importance of contextual understanding in multimodal models, as demonstrated by Claude 3 and GPT-4 Turbo.

### A.6.2   TASK II: EMBODIED QUESTION ANSWERING

As presented in Figure 11, fuyu-8B responded by identifying three traffic lights visible in the images. However, this response is inaccurate according to the ground truth, which states that only two traffic lights are present. This overestimation indicates a potential issue with embodied recognition or differentiation in Fuyu-8B.

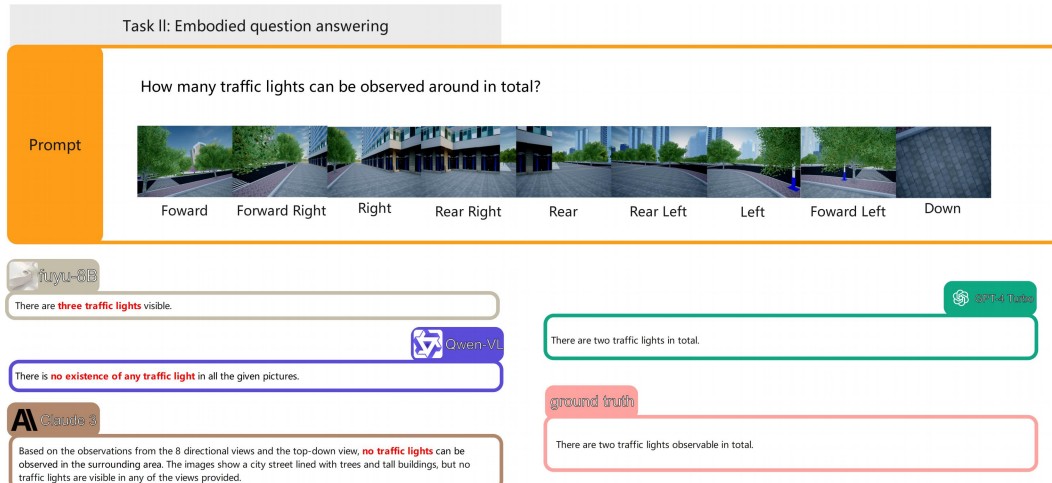

Figure 11: This case of embodied question answering task involves answering the question "How many traffic lights can be observed around in total?" based on images from eight directions (forward, forward-right, right, rear-right, rear, rear-left, left, forward-left) and one top-down view. The original outputs of different models are listed separately.

Qwen-VL asserted that there are no traffic lights visible in any of the provided images. This response is also incorrect, as it fails to recognize the two traffic lights that are present. This suggests a limitation in Qwen-VL's ability to detect specific objects accurately in a multimodal context.

Claude 3 similarly concluded that there are no traffic lights observable in the images. This response, like that of Qwen-VL, indicates a failure in object detection capabilities, as it overlooks the traffic lights that are present.

GPT-4 Turbo, on the other hand, correctly identified that there are two traffic lights in total. This response aligns with the ground truth, demonstrating GPT-4 Turbo's superior ability to accurately recognize and count specific objects within the provided visual context.

The accuracy of the responses varies significantly among the models. GPT-4 Turbo stands out as the only model to provide the correct answer, reflecting its strong performance in visual recognition and comprehension tasks. In contrast, Fuyu-8B overestimates the number of traffic lights, while Qwen-VL and Claude 3 both fail to recognize any traffic lights, highlighting potential limitations in their object detection capabilities.

### A.6.3  TASK III: EMBODIED DIALOGUE

The models' performances are compared to the ground truth to evaluate their accuracy and descriptive capabilities, as shown in Figure 12. Fuyu-8B acknowledges the presence of prominent waypoints but incorrectly identifies their nature and location. It states that the waypoints are located at a stadium on a tiled floor, likely inside a sports facility. This response is inaccurate as it fails to mention the actual prominent waypoints, such as distinctive buildings or structures, and does not align with the ground truth. Qwen-VL identifies an area of trees that can be used for navigation but fails to recognize the prominent architectural structures. It provides a detailed list of all image directions but does not accurately describe the waypoints. This response is partially correct in identifying natural features but misses the crucial architectural landmarks highlighted in the ground truth. Claude 3 asserts that surrounding skyscrapers and tall buildings serve as prominent waypoints or landmarks in the urban environment. It accurately pinpoints the locations of these landmarks, noting their visibility in the south and southwest directions relative to the current position. This response closely aligns with the ground truth, demonstrating a good understanding of the urban landscape and identifying the correct waypoints. GPT-4 Turbo provides a precise and accurate description, identifying a prominent structure resembling twin towers with unique architectural design as major waypoints. It specifies that these towers are visible in the rear-left direction. This response most accurately reflects the ground truth, which mentions distinctive buildings and tree-lined walkways as prominent waypoints.

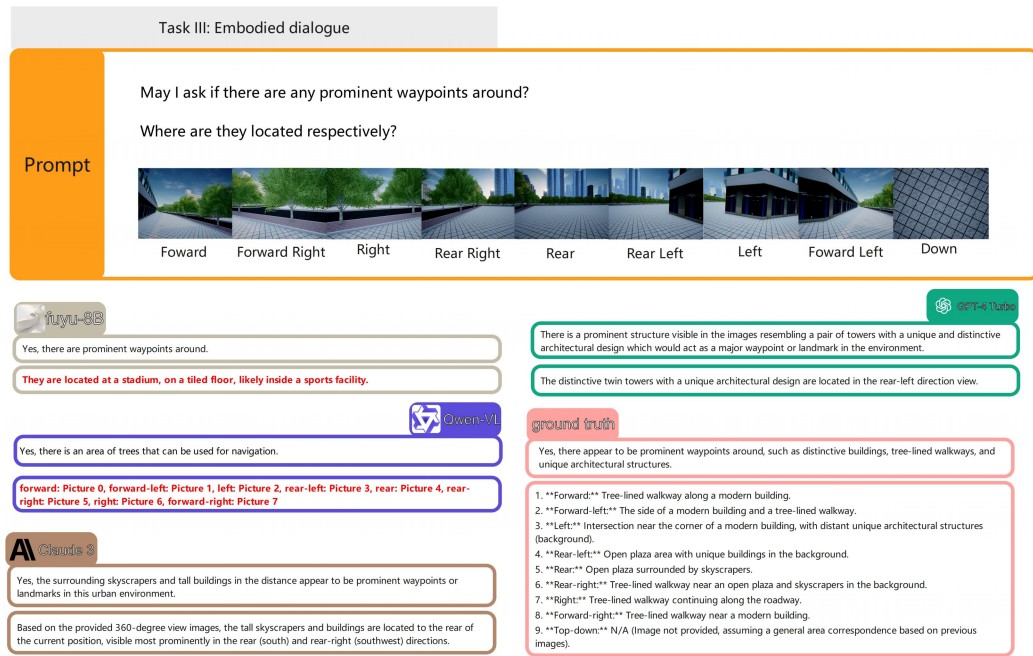

Figure 12: This embodied dialogue case involves a dialogue where the models are asked to identify any prominent waypoints around and specify their locations based on a series of images taken from eight different directions (forward, forward-right, right, rear-right, rear, rear-left, left, forward-left) and one top-down view.

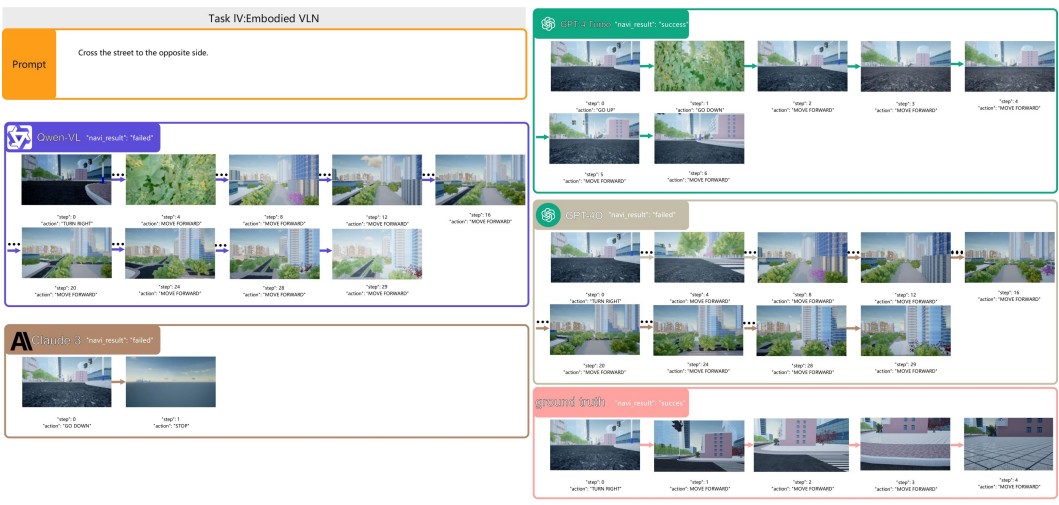

Figure 13: The agent should decide the action according to the observations until arriving the goal.

Among the models, GPT-4 Turbo provides the most accurate and descriptive response, closely aligning with the ground truth by identifying the twin towers as prominent waypoints. Claude 3 also offers a strong response by correctly identifying the surrounding skyscrapers and their specific locations. In contrast, Fuyu-8B and Qwen-VL fail to accurately identify the architectural landmarks, highlighting the need for improvement in their embodied ability to recognize and describe complex urban environments.

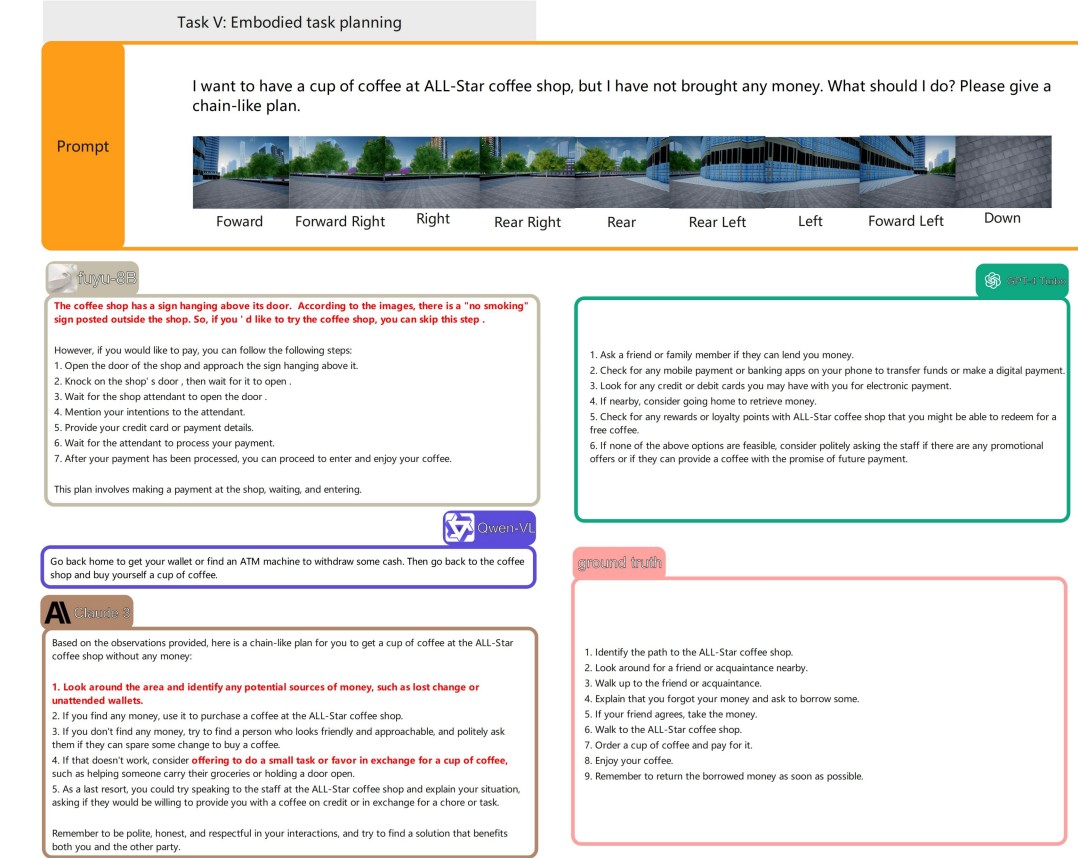

Figure 14: This case of embodied task planning involves creating a chain-link plan to get a cup of coffee from the ALL-Star coffee shop without having brought any money. The AI models are asked to provide a step-by-step plan based on a series of images taken from eight different directions (forward, forward-right, right, rear-right, rear, rear-left, left, forward-left) and one top-down view.

### A.6.4 TASK IV: EMBODIED VLN

In order to compare different models on the VLN task, we give a detail case in Figure 13. The analysis reveals that only GPT-4 Turbo successfully completes the task, suggesting it has a superior capability in interpreting and navigating based on RGB observations. Both Qwen-VL and GPT-4o show similar patterns of failure, indicating potential areas for improvement in their navigation algorithms. Claude 3's failure highlights a critical need for enhancement in its initial perception and decision-making processes. The ground truth provides a clear and effective navigation path, demonstrating the importance of precise and context-aware actions in achieving the objective.

### A.6.5 TASK V: EMBODIED TASK PLANNING

As shown in Figure 14, Fuyu-8B's response focuses on a detailed description of the coffee shop, mentioning a "no smoking" sign. It then provides a procedure involving opening the door, waiting for the shop to open, mentioning intentions to the attendant, and providing payment details. This plan is not practical as it assumes the user has money or a payment method, which contradicts the prompt's condition of not having brought any money. Qwen-VL suggests going back home to get money or finding an ATM to withdraw cash before returning to the coffee shop. While this response is practical, it lacks creativity and does not explore alternative solutions available in the immediate environment, making it less optimal than the ground truth. Claude 3 provides a detailed and creative plan, which is practical, creative, and aligns well with the ground truth, addressing the situation effectively without requiring the user to leave the area. Similar to Claude 3, GPT-4 Turbo's response is practical and

creative, providing several feasible options without needing to leave the vicinity, and aligns well with the ground truth.

Claude 3 and GPT-4 Turbo provide the most practical and creative solutions, closely aligning with the ground truth. They explore multiple options to solve the problem without requiring the user to leave the immediate area. Fuyu-8B's response is less practical as it does not address the lack of money, and Qwen-VL's solution, while practical, lacks creativity and does not leverage immediate resources.

