# OpenReview forum: "EmbodiedCity: A Benchmark Platform for Embodied Agent in Real-world City Environment"
_ICLR.cc/2025/Conference — Submitted to ICLR 2025_

### Official Review · Reviewer_hGNu · 2024-11-03

**Soundness:** 2
**Presentation:** 2
**Contribution:** 2
**Rating:** 5
**Confidence:** 3

**Summary:**

The paper presents a benchmark platform for evaluating embodied artificial intelligence in realistic urban environments, addressing gaps in open-world scenarios. It features a detailed 3D simulation, diverse evaluation tasks, and user-friendly interfaces, enhancing embodied intelligence capabilities and supporting practical applications in artificial general intelligence.

**Strengths:**

1. The paper constructs a detailed 3D environment based on real-world urban settings in Beijing, improving on previous fictional models.
2. The paper establishes a diverse set of evaluation tasks that assess various dimensions of embodied intelligence.
3. The paper provides accessible input and output interfaces for easy interaction and performance evaluation of embodied agents.

**Weaknesses:**

1. The motivation behind this paper aligns with the principles of ELM [1], focusing on embodied understanding in driving scenarios. A detailed explanation of the differences between the two approaches is necessary.
2. Most of the evaluation tasks already exist in current literature. Providing a detailed explanation to distinguish these tasks from those in other works is important.

[1] Embodied Understanding of Driving Scenarios

**Questions:**

Please refer to the weaknesses section.

---

### Official Review · Reviewer_XRCg · 2024-11-03

**Soundness:** 3
**Presentation:** 2
**Contribution:** 1
**Rating:** 3
**Confidence:** 4

**Summary:**

This paper proposed an open-world simulator for embodied agents. The simulator is based on the city Beijing. To evaluate agents in this simulator, the authors propose 5 tasks. Embodied Scene Understanding, Embodied Question Answering, Embodied Dialogue, Embodied action (navigation), and Embodied Task Planning.

They evaluate 4 current VLMs on these tasks.

**Strengths:**

The proposed simulator and environment covers a large area.

The authors create various tasks in the simulator.

The authors evaluate various current VLMs on their proposed tasks.

**Weaknesses:**

Visuals. The paper advertises high quality visuals, and rates their visuals 3 out of 3 stars. To the reviewers, the visuals do not look better than things rated 2 out of 3 stars, such as CARLA.

Evaluation metrics. Evaluating Embodied QA, Embodied Dialogue, and Embodied Task Planning with captioning and translation metrics, BLUE, CIDEr, etc, seems like a poor choice. I encourage the authors to define a notion of success for each task that evaluates if the agent did the task correctly. Such as, for the Embodied QA and Dialogue tasks, making questions with ambiguous answers multiple choice, or using something like LLM-Match (https://open-eqa.github.io). Questions without ambiguous answers can be evaluated directly. This would lead to a more meaningful and interpretable metric.

Missing References. This paper is missing a very large number of references. For example, the authors mention, by name, the tasks Vision-and-Language Navigation (VLN) (https://arxiv.org/abs/1711.07280) and Embodied QA (https://arxiv.org/abs/1711.11543), but do not cite either work. They also do not cite the paper that proposed SPL (https://arxiv.org/abs/1807.06757). Overall, the space of EmbodiedAI has seen considerable interest and work but the paper cites very little of the work in this area.

**Questions:**

How much navigation, if any, is required for the Embodied QA, Embodied Dialogue, and Embodied Task Planning tasks?

What are the mean lengths of the "Short" and "Long" paths for the VLN task?

What is the performance of the simulator like?

---

### Official Review · Reviewer_eq7u · 2024-11-04

**Soundness:** 2
**Presentation:** 3
**Contribution:** 2
**Rating:** 3
**Confidence:** 5

**Summary:**

In this paper, the authors construct a benchmark platform for embodied intelligence evaluation in real-world city environments. They create a highly realistic 3D simulation environment based on real city elements and conduct high-fidelity simulations of pedestrian and vehicle flows. The platform has a set of evaluation tasks and provides input and output interfaces. The quantitative evaluation is performed over popular large language models on this platform.

**Strengths:**

1. The authors introduce a new urban simulator for simulating pedestrians and traffic states of a city.
2. This work provides the resources of a large digital city district, which is quite scarce in this field.
3. This study evaluates several state-of-the-art large multimodal models (LMMs) against the proposed benchmark to assess their effectiveness in addressing embodied tasks from multiple perspectives. The results largely align with findings from other LMM benchmarks, which partially support the validity of the proposed benchmark.

**Weaknesses:**

1. Some metrics presented in Table 1 appear to be subjective and potentially incorrect. For instance, regarding visual realism, the rendering quality in Figure 1 is noticeably less convincing compared to GRUtopia. The images appear to be produced by a rasterization renderer rather than a ray tracing or path tracing renderer, revealing a significant disparity between the quality of human-crafted assets and actual buildings. Furthermore, from an embodiment perspective, the platform seems to primarily incorporate drones and vehicles, lacking support for widely-used embodiments such as humanoid and quadruped robots, despite the authors' claim in Table 1 that all these embodiments are supported.
2. The diversity of the QA templates illustrated in Figures 8 and 9 appears to be quite limited. A broader range of templates would enhance the comprehensiveness of the evaluation.
3. While the authors assert that the scene is crafted from real city maps, they do not clarify the benefits of this approach. The quality of the assets and rendered images does not seem realistic enough to justify this claim. Additionally, the authors have not demonstrated the sim-to-real potential of the proposed dataset, which is crucial for its application.
4. Although the report includes scores based on several metrics, there is a lack of intuitive illustrations to showcase what the large multimodal models (LMM)-agents excel at solving. The results presented do not clearly reveal the main challenges of the proposed tasks.
5. The rationale for incorporating dynamic pedestrians and vehicles into this platform is not clearly articulated. There appears to be no strong connection between the proposed tasks and the roles of pedestrians and vehicles, which raises questions about their necessity in the framework.
6. Details regarding the LMM agents are insufficiently described. It remains unclear how these agents handle sequential egocentric observations, which is essential for understanding their operational effectiveness.
7. The usefulness of the proposed benchmark is not adequately established. The absence of learnable baselines to validate the dataset’s rationale potentially limits the significance and impact of this work.
8. The authors do not justify the running efficiency of the platform, which is critical for scaling training within the environment. A discussion of performance metrics or benchmarks would be beneficial.
9. The authors have not conducted experiments to explore the impact of different embodiments on task performance. Such investigations could provide valuable insights into the effectiveness of various embodiment strategies.
10. The metrics for Evaluative Question Answering (EQA) rely on conventional reference-based NLP metrics, which may not directly demonstrate the correctness of the answers provided. It would be more effective for the authors to utilize a large language model (LLM) to assess the correctness of answers in relation to the ground truth.

Typos:
1. In the caption of Table 6, "vision-and-navigation" should be corrected to "vision-and-language navigation."

**Questions:**

See weaknesses.

---

### Official Review · Reviewer_RW8a · 2024-11-04

**Soundness:** 2
**Presentation:** 2
**Contribution:** 2
**Rating:** 3
**Confidence:** 4

**Summary:**

This paper presents a comprehensive benchmark platform aimed at assessing the performance of embodied agents in a realistic urban setting. Unlike previous benchmarks limited to indoor or fictional settings, this platform features a highly realistic 3D simulation of an actual city district in Beijing. The benchmark includes five core tasks for evaluating embodied capabilities: scene understanding, question answering, dialogue, visual language navigation, and task planning. These tasks are designed to capture the core embodied AI abilities of perception, reasoning, and decision-making. The platform supports multiple agents, offers an interface for real-time control, and provides a SDK for easy access, along with a dataset for training and evaluation.

**Strengths:**

- The platform's integration with Unreal Engine and AirSim, along with the provision of a Python SDK, significantly lowers the barrier for use and promotes flexible, scalable experimentation for researchers.
- The benchmark includes evaluations of popular large language models (e.g., GPT-4, Claude 3) across tasks, providing a well-rounded quantitative baseline for the embodied intelligence community.
- The open structure allows future expansions, such as multi-agent collaboration and adaptability, fostering an extensible environment for advanced research in embodied AI.

**Weaknesses:**

1. While the paper addresses the city layout aspect of the sim-to-real gap, it does not extend to other critical factors impacting real-world applicability. Additionally, no experiments are conducted to quantify the sim-to-real benefits derived from using a real-world city layout, leaving the practical advantages of this choice unclear.
2. The shadows and lighting in Figure 3 appear less realistic, which may limit the benchmark's effectiveness in simulating real-world visual conditions.
3. The benchmark predominantly focuses on drone-related tasks, with limited discussion on tasks relevant to autonomous vehicle planning. Definitions, metrics, and methodologies for evaluating embodied tasks in autonomous driving contexts, particularly for planning, are not included.
4. The tasks are largely oriented toward language-based interactions, with an emphasis on using large language models. Metrics like BLEU and ROUGE, which primarily measure text quality, may not fully capture the performance of embodied AI tasks, raising questions about the suitability of these metrics for this benchmark.
5. The paper does not specify a license for the assets used. Given that some assets are sourced from Unreal Engine, Baidu Maps, and Amap, it remains unclear whether these assets are freely distributable under their original licenses. Clarification on the licensing terms for these assets would strengthen the transparency and accessibility of the benchmark.

**Questions:**

See weaknesses

---

### Meta-Review · Area_Chair_DxBo · 2024-12-23

**Metareview:**

This paper was reviewed by four field experts and received unanimously negative evaluations. The main concerns raised include a lack of significant technical contributions and relatively underwhelming results. Additionally, no rebuttal was provided by the authors. The AC finds no reason to recommend acceptance.

**Additional Comments On Reviewer Discussion:**

No rebuttal was submitted.

---

### Decision · Program_Chairs · 2025-01-22

Reject